# Discovering the drivers of clonal hematopoiesis

Oriol Pich [1,4], Iker Reyes-Salazar[1], Abel Gonzalez-Perez [1,2 ✉] & Nuria Lopez-Bigas [1,2,3 ✉]

Mutations in genes that confer a selective advantage to hematopoietic stem cells (HSCs) drive clonal hematopoiesis (CH). While some CH drivers have been identified, the compendium of all genes able to drive CH upon mutations in HSCs remains incomplete. Exploiting signals of positive selection in blood somatic mutations may be an effective way to identify CH driver genes, analogously to cancer. Using the tumor sample in blood/tumor pairs as reference, we identify blood somatic mutations across more than 12,000 donors from two large cancer genomics cohorts. The application of IntOGen, a driver discovery pipeline, to both cohorts, and more than 24,000 targeted sequenced samples yields a list of close to 70 genes with signals of positive selection in CH, available at http://www.intogen.org/ch. This approach recovers known CH genes, and discovers other candidates.

[1] Institute for Research in Biomedicine (IRB Barcelona), The Barcelona Institute of Science and Technology, Baldiri Reixac, 10, 08028 Barcelona, Spain. [2] Research Program on Biomedical Informatics, Universitat Pompeu Fabra, Barcelona, Catalonia, Spain. [3] Institució Catalana de Recerca i Estudis Avançats (ICREA), Barcelona, Spain. [4] Present address: Cancer Evolution and Genome Instability Laboratory, The Francis Crick Institute, London, UK. ✉email: abel.gonzalez@irbbarcelona.org; nuria.lopez@irbbarcelona.org

Clonal hematopoiesis (CH) is a condition related to aging across the human population[1–9], driven by somatic alterations that appear in hematopoietic stem cells (HSCs) and confer them a selective advantage. It was recognized through cytogenetic studies in the 1960s and its genetic bases and prevalence with aging were first discovered through studies of non-random X chromosome inactivation in women[10,11]. In recent decades, genomic studies of thousands of donors without any hematologic phenotype identified CH causal somatic variants in genes known to be associated to hematopoietic malignancies, such as DNMT3A, TET2, ASXL1, TP53, JAK2 and SF3B1[1,2,8,10,12–15]. The progeny of an HSC bearing mutations of one of these genes develops in a process of clonal expansion. The presence of CH, in turn, is known to be associated with other health risks, such as the development of hematopoietic malignancies or increased incidence of cardiovascular disease[2,4,5,7,16].

The aforementioned human genomic studies, and more recent analyses[16–22] have identified a list of CH-causing somatic variants. Nevertheless, their identification is hampered by the fact that the clonal expansion related to CH is rather modest, and therefore, it presents with low variant allele frequency (VAF). This has determined the development of two main strategies of detection of CH drivers. On the one hand, some projects with access to deep sequencing data of particular sites of the human genome (e.g., tumor panel sequencing) identify CH with exquisite sensitivity, but only if the causing variant overlaps with the sites sequenced[14,16–18,20]. On the other hand, whole-genome or whole-exome sequencing data has been exploited to identify blood somatic variants exploring the region of VAF below the one corresponding to germline variants[13,21–23]. This approach is thus only able to detect relatively large CH clones. One important caveat of both approaches is that not all genes affected by mutations across blood samples (even known cancer driver genes) are drivers of CH. Whereas sequencing more blood samples will lead to the identification of more recurrently mutated suspicious genes, many of them are prone to be passengers of this clonal expansion process.

Thus, an accurate and complete list of CH-related genes remains elusive to date. Completing it is essential to comprehensively identify CH in individuals, to ascertain their risk to develop related diseases and to complete our knowledge of the molecular mechanisms underlying CH.

In recent years efforts to identify genes with mutations under positive selection in tumorigenesis have begun to uncover the compendium of mutational cancer driver genes[24–27]. Since the clonal expansion that drives CH is reminiscent of that observed in tumors, methods to detect positive selection in the mutations of genes across tumors may be applied to identify the complete list of CH-related genes. Detecting these signals of positive selection depends on an accurate identification of blood somatic mutations.

Here, we repurpose blood and tumor samples of donors with no known hematopoietic malignancy obtained from primary[28] (N~8,000) and metastatic[29] (N~4000) cancer genomics initiatives to detect somatic mutations in blood. To this end, we use the paired tumor sample as the reference germline genome of the donors in these two cohorts. On the set of blood somatic mutations identified in these two cohorts and across 24,146 other targeted sequenced tumors, we then run the Integrative Onco-Genomics (IntOGen[25]) pipeline that implements seven state-of-the-art driver discovery methods. As a result, known CH-related genes and other genes with no previous report of association with CH are identified. Our results serve as a proof of concept of the validity of this strategy and open up the opportunity to repurpose cancer genomics data in the public domain to identify the compendium of CH driver genes, of which this paper presents a snapshot.

## Results

**Identifying somatic mutations in blood samples.** We reasoned that low-coverage whole-genome sequencing of blood samples routinely carried out in cancer genomics projects may be repurposed to detect CH. To this end, we obtained the DNA sequences of blood and tumor samples (paired samples) from two large cancer cohorts. The first cohort comprised 3785 paired samples obtained from metastatic solid cancer patients (metastasis cohort) sequenced at the whole-genome level[29]. The second included 8530 paired samples collected from primary solid tumor patients (primary cohort) sequenced at the whole-exome level[28]. In both cohorts, we focused only on donors with solid tumors because in hematopoietic malignancies a full clonal expansion associated with the cancer is present in the blood sample.

Although possible, the identification of somatic mutations in the blood samples taken from the donors of these cohorts is extremely challenging due to the low coverage employed to sequence them. In this scenario, subclonal mutations are hard to distinguish from random sequencing errors. Moreover, germline variants may be falsely called somatic if a somatic mutations calling is carried out on the blood sample alone. We reasoned that this problem could be overcome using the second (tumor) sample taken from the same patient as a reference of their germline genome. A comparison of the variants identified in the blood sample and the tumor sample with respect to the human reference genome would then reveal the somatic mutations specific to hematopoietic cells (Fig. 1a).

We thus–inspired by a previous approach to identify early mutations in the development of the hematopoietic system[30]–implemented a pipeline to systematically carry out this "reverse" somatic mutation calling on the paired samples of the two cohorts (Fig. 1b; Supp. Figure 1a; Supp. Note 1). First, blood mutations are identified using a somatic mutation caller widely employed in cancer genomics studies[31], and a set of filters are applied to guarantee that these are true somatic mutations rather than germline variants or random sequencing errors. In the metastasis cohort, this yields ~1 million candidate whole-genome somatic mutations across 3785 blood samples. We call this the full catalog of somatic mutations. Two further filtered sets are obtained applying one of two criteria (Fig. 1b): mutations also identified by a second widely-employed somatic caller[32] (mutect catalog), or mutations also identified as likely somatic by MosaicForecast, an algorithm trained for this task using phased mutations[33] (mosaic catalog; Supp. Figure 1b). Importantly, the reverse calling approach empowers the detection of variants in known CH genes at values of VAF unattainable by a typical calling of germline variants (hereafter, germline calling) on a single whole-genome sequenced blood sample (Fig. 1c). Taking as example three well known CH drivers (TET2, DNMT3A and ASXL1), more than 30% of all mutations identified by the reverse calling are missed by a germline calling. Across 15 well-known CH driver genes[34] 37% of all variants are identified by the reverse calling, but missed by the germline calling. More importantly, 91% (1566 out of 1714) of all variants that the germline calling would identify across these genes are likely not somatic, as evidenced by the fact that they are not identified by the reverse calling (Fig. 1c). These two comparisons highlight that the reverse calling attains higher sensitivity and specificity in the identification of blood somatic mutations than the germline calling.

**Somatic mutations in blood samples show evidences of clonal hematopoiesis.** Only variants shared by enough blood cells–those that derive from the clonal expansion underlying CH– are expected to appear above the limit of detection of the sequencing. Therefore, the detection of these somatic mutations through the

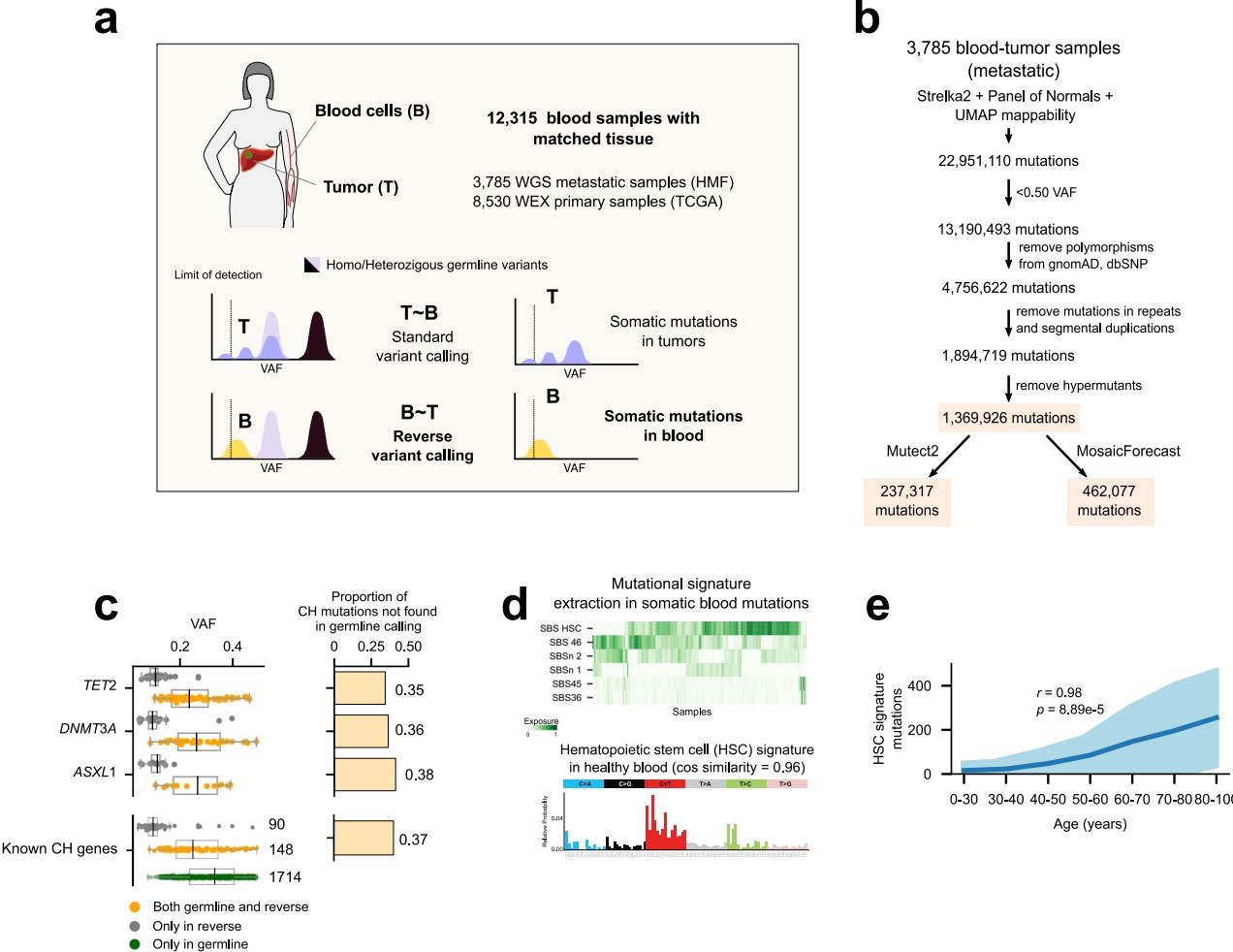

**Fig. 1 The reverse calling approach to detecting blood somatic mutations. a** Somatic mutations in blood are identified by comparing variants in the blood/tumor paired samples from a cancer patient. We applied this approach to two cohorts of primary and metastasis tumors totalling 12,315 blood donors with no known hematologic malignancy. **b** Flowchart of the reverse calling and filtering approach. Numbers correspond to mutations remaining in the dataset of the metastasis cohort (full, mosaic or mutect) after each filtering step. **c** Somatic mutations identified by the reverse calling and a one-sample germline variant calling across blood samples in the metastasis cohort ($N = 3,785$). Boxplots represent the distribution of VAF of variants affecting well-known CH driver genes identified only by the reverse calling (gray), by both approaches (yellow) or only by the germline calling (green). In the boxplots, the box represents the second and third quartiles, separated by a line indicating the median; the whiskers represent the minimum and maximum of the distribution excluding outliers. Right-hand barplots illustrate the fraction of mutations affecting each gene that are identified only by the reverse calling approach. **d** Top, activity of mutational signatures in the blood samples of donors across the metastasis cohort ($N = 3,785$) identified using the mosaic set; bottom, mutational profile of tri-nucleotide probabilities of one of the signatures extracted from the cohort which highly resembles (cosine similarity = 0.96) that of a signature active in healthy hematopoietic stem cells (HSCs). **e** Relationship between the number of mutations contributed by the HSC signature across blood samples in the metastasis cohort and the (binned) age of their donors. The mean activity of the signature across donors of each bin is represented as the dark blue line, with its standard deviation in light blue color. A significant positive correlation between the two variables is apparent. The p-value corresponds to the Pearson's regression coefficient. WGS whole genome sequencing, HMF metastasis cohort, TCGA primary cohort, WEX whole exome sequencing, VAF variant allele frequency, CH clonal hematopoiesis, SBS single base substitution, HSC hematopoietic stem cell, cos cosine. Source data for panels **c**, **d** and **e** are provided as Source Data files.

reverse calling approach is an evidence of CH in the samples of both cohorts (Fig. 1b and Supp. Figure 1a).

We expect that these mutations exhibit a tri-nucleotide profile characteristic of variants spontaneously appearing as HSCs divide[35]. The identification of mutational signatures active in the blood samples of the metastatic cohort yielded 6 distinct profiles. Some of these are similar to signatures previously associated with sequencing artifacts[36] (Supp. Figure 1c, d; Supp. Data file 1). Nevertheless, the most pervasive mutational signature in the cohort shows a profile that is virtually identical (cosine similarity = 0.96) to that of the known hematopoiesis signature (Fig. 1d). This constitutes further evidence that a set of somatic mutations contributed by hematopoiesis are present

across these healthy blood samples. Moreover, it is further indication that CH is present across at least some of the donors.

We also expect that blood somatic mutations contributed by HSC divisions increase with the age of the donors[35,37]. First, the chance of appearance of a CH mutation (a mutation affecting a CH driver), and in consequence the chance of the expansion of a HSC clone, increases with age. Second, the number of hematopoietic mutations in this HSC clone founder (which become amplified due to the clonal expansion), also increases with age, because hematopoietic mutations are acquired at a steady rate with every HSC division. Third, the longer the time elapsed between the beginning of the clonal expansion and the obtention of the sample (which naturally increases with the

donor's age), the higher the VAF of the hematopoietic mutations, and the likelihood that they rise above the limit of detection of bulk sequencing. In agreement with this expectation, we observed that the number of hematopoiesis mutations identified in the metastasis cohort applying the reverse calling approach increases with the age of the donor (Fig. 1e; Supp. Figure 1d illustrates the relationship for all phased mutations). On the contrary, the number of mutations contributed by the other signatures extracted from the cohort does not increase steadily with age (Supp. Figure 1e).

In summary, several lines of evidence provide support to the reverse calling approach as an efficient method to identify somatic mutations in blood samples of patients with CH when a paired tissue sample is available.

**Discovery of clonal hematopoiesis drivers.** We reasoned that, as is the case in the clonal expansion related to tumorigenesis[25,38], the mutational patterns of CH-associated genes should exhibit signals of positive selection across donor blood samples. Therefore, methods that have been developed to identify these signals of positive selection in cancer[25,38–41] could be applied to somatic mutations in blood samples to identify the genes with significant deviations from their expected patterns of mutations under neutrality. Anchored in these methods, cancer genomics researchers have set the goal of uncovering the compendium of cancer driver genes. Analogously, exploiting these methods empowers us to open a roadmap to the compendium of CH driver genes.

To test this concept, we applied the IntOGen pipeline[25] (which runs seven state-of-the-art driver discovery methods[42–48] and combines their results) to blood somatic mutations in the primary and metastatic cohorts (Fig. 2a). We filtered out blood somatic

mutations with VAF above 0.4, to minimize the risk that falsely called mutations enter the discovery process (Supp. Note 1). Each of these methods identifies one or more signals of positive selection (e.g., abnormally high recurrence of mutations, unexpected clustering of mutations in certain regions of the gene, or exceptionally high functional impact of the observed mutations) in the mutational pattern of genes across samples (Fig. 2a and Supp. Note 1). False positive genes identified by a particular method are filtered out by the combination of their outputs through a voting-based approach[25]. Finally, 15 genes that are significant according to the combination approach are filtered out as they are deemed suspicious after a careful vetting that considers gene expression across HSCs, somatic hypermutation processes, common sequencing artifacts and frequent false positive genes of the driver discovery process (Supp. Note 1). We also applied the IntOGen pipeline to the somatic mutations identified across 24,146 targeted-sequenced paired blood/tumor samples[17,49] (targeted cohort) in which a mutation calling filtering variants in common with the tumor sample was carried out (Supp. Note 1).

The lists of CH drivers are composed of 26 genes identified in the metastasis cohort, 21 genes from the primary cohort, and 43 in the targeted cohort (Fig. 2b; Supp. Data files 1 and 2). All fifteen well-known CH-related genes, obtained from Fuster et al.[34] (CH known drivers) are identified. Validation of the involvement of 26 other genes (such as *ATM* and *CHEK2*) comes from the fact that they have been identified as drivers of hematopoietic malignancies[17] (Fig. 2b). We did not find previous reports of involvement of the remaining 23 genes, some of which (e.g., *ABL2*, *FOXP1* and *TP63*) are known cancer drivers[50], in CH. Nevertheless, several lines of evidence gathered across the literature (summarized in Supp. Data file 2) support the involvement of the majority of them in CH. We –as others

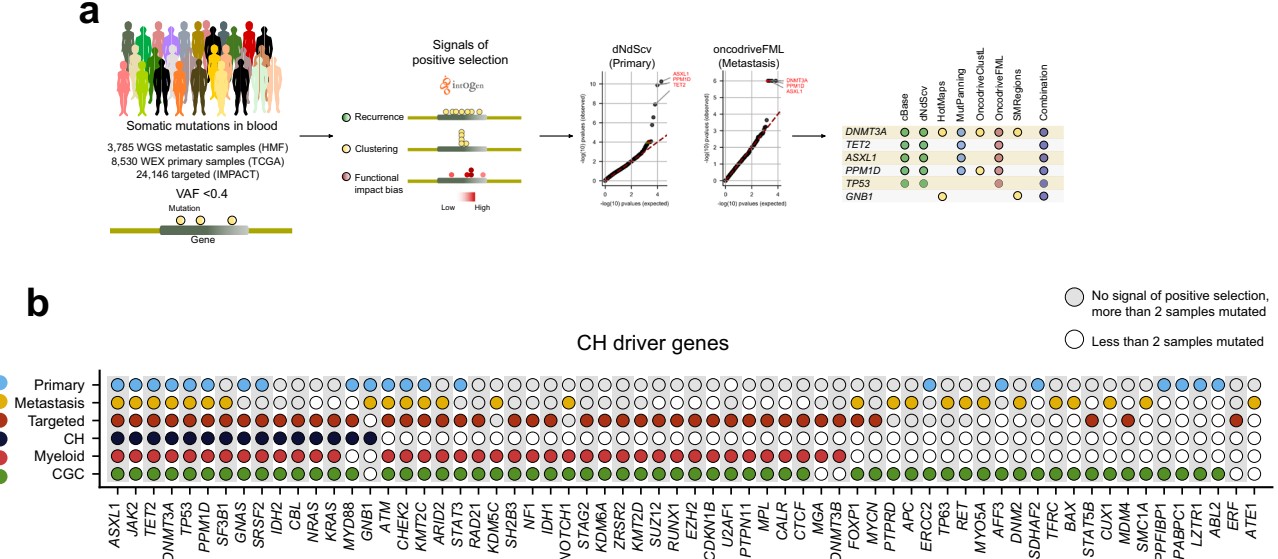

**Fig. 2 Discovery of clonal hematopoiesis driver genes. a** Summary of the discovery analysis applied to blood somatic mutations detected across primary, metastasis and targeted cohorts. The (differently filtered) sets of blood somatic mutations identified across all donors of a cohort were the input data for the analysis. Seven state-of-the-art driver discovery methods probing different signals of positive selection were applied (via the IntOGen pipeline) to each dataset of mutations. The distribution of expected and observed p-values (qq plots) for two of these methods (which implement parametric, non-parametric or empirical statistical tests described in the corresponding original articles) are represented in the panel. The IntOGen pipeline also handles the combination of the output of the seven methods to yield a unified list of CH driver genes in each cohort (details in Supp. Note 1). **b** CH driver genes discovered across the three cohorts. Genes known to be involved in CH, myeloid malignancies or tumorigenesis in general are labeled with different colors (denoted at the left of the plot). The union of the lists of CH drivers discovered in these three cohorts (64 genes) integrate the CH drivers compendium presented in Supplementary Data file 2 and available through www.intogen.org/ch. IMPACT: targeted cohort, CGC cancer gene census. Source data for panel b are provided as Source Data files.

before– observe an important overlap between CH drivers and known cancer drivers. Mutations affecting these genes and conferring mutant HSCs a growth davantage are likely to be under positive selection in CH development, similarly to their role in tumorigenesis. While much less is known of the potential role of purifying selection in the evolution of CH, a recent report suggests that it is probably not negligible[51].

In summary, the identification of signals of positive selection in the pattern of somatic mutations of the genes across blood samples of individuals without hematologic disease is an effective way to discover CH-related genes, it recovers most known CH genes and has the power to discover others. This compendium–the snapshot presented in this work–comprises the genes identified across the primary, the metastasis and the targeted cohorts and is available in Supplementary Data file 2 and through https://www.intogen.org/ch.

**The drivers of clonal hematopoiesis**. To characterize the discovered CH-related genes, we probed the association of their mutations with several physiological and clinical variables relevant to the development of CH (Fig. 3a, b). As previously reported[17], across patients in the metastasis cohort, we found that the emergence of CH is positively influenced by age and by the exposure to cytotoxic (but not non-cytotoxic) anticancer treatments (Fig. 3a). This positive association with age is maintained when mutations in CH-related genes that are not in the list of well-known CH drivers across the primary cohort are analyzed as a group (Supp. Figure 2a)–and some of them, individually–, supporting the involvement of these genes in the development of age-related CH. We also recapitulated the prior knowledge that mutations in certain genes, such as PPM1D and CHEK2 are positively associated with prior exposure to platinum-based drugs (Fig. 3b). Indeed, mutations in a group of three DNA-damage response genes (TP53, PPM1D, CHEK2) appear significantly associated with the exposure to platinum (Fig. 3b), probably because HSCs carrying them possess a better chance at survival than others when exposed to these DNA-damaging chemotherapeutics[2]. When the representation of cancer types across donors in the primary cohort is taken into consideration, a strong significant relationship between thymomas and CH cases is apparent. This could be related with the appearance of auto-immunity mediated by the clonal expansion of T-cells that is observed in thymomas[52]. A weaker negative association with cases of breast and bladder cancer is also observed. In any event, the detection of CH showed no significant association with the majority of malignancies represented in the primary cohort (Supp. Figure 2b), indicating that CH frequency in this cohort likely reflects the underlying risk of CH in the general population. There are no apparent differences in the distribution of VAF of the somatic mutations affecting known CH drivers, known myeloid drivers and other putative CH drivers across the primary and metastasis cohorts (Supp. Figure 2c).

We then asked whether the pattern of CH-related mutations of known cancer genes differ from that of their oncogenic mutations (Fig. 3c and Supp. Figure 3a). In the case of DNMT3A, one of the main hotspots of CH-related mutations (affecting residue 882) also appears recurrently mutated across tumors, while two other hotspots (residue 635 and 736) seem to be more specific to CH. In the case of TP53 mutations in both CH and cancer cases appear clustered within the DNA binding domain. The distribution of mutations of PPM1D is very similar across CH and cancer cases. In both scenarios, PPM1D truncating mutations close to the C-terminal yield a protein product lacking a degron, which is thus abnormally stable and results in the down-regulation of DNA-damage response and the proliferation of cells in the presence of

such damage[53]. Mutations across CH and cancer cases are also very similarly distributed along the gene in the case of MYD88 (with one dominant hotspot), but their distributions differ in IDH2. The pattern of mutations observed in these CH genes across the primary and metastasis cohorts resembles those obtained across the targeted cohort (Supp. Figure 4a). The distribution of mutations along the sequence of other genes in the compendium is shown in Supplementary Figure 3a.

While many CH drivers exhibit similar frequency of truncating mutations across both CH and myeloid cancer cases, in some, a clear enrichment (TET2, PPM1D) or depletion (NOTCH1, ARID2) of truncating mutations is observed across CH (Fig. 3d). Interestingly, the rate of truncating mutations in CH driver genes across donors of the primary and metastatic cohorts is very similar to that observed in the targeted cohort (Supp. Figure 4b). The case of NOTCH1, mutations of which are related with the development of hematopoietic malignancies, such as ALL and CLL, could indicate that different selective constraints underlie the development of CH and these malignancies[54]. (The low share of truncating mutations of NOTCH1 is observed across the three cohorts analyzed; Supp. Figure 4b.) Overall, the observed differences between CH and cancer may have their origin not only in different evolutionary constraints in the development of both processes, but also in the disparate array of mutational processes active in healthy blood and tumors.

**Detecting clonal hematopoiesis across ~12,000 donors**. We then identified all patients across the primary and metastasis cohorts with a potential protein-affecting somatic mutation in one or more genes in the compendium of CH drivers (Fig. 4a). The rate of mutations of the most frequently mutated CH genes varies between both cohorts (Fig. 4b), likely reflecting differences in mutational processes and evolutionary constraints related to CH emergence. The most frequent mutational hotspots affect JAK2 and DNMT3A (Fig. 4c). Interestingly, while more than three-quarters of the patients with mutations affecting CH drivers across both cohorts present only one mutation affecting a CH gene, more than one are identified in 18% (Fig. 4d). These co-occurring mutations affect some CH-related genes more frequently (Fig. 4e; Supp. Fig. 5a) with FOXP1, SF3B1 and PPM1D among the genes with most frequent co-mutations.

The range of VAF of the mutations in these genes reveals a wide spectrum of clonal expansion across CH cases (Supp. Fig. 5b). The rate of hematopoiesis mutations –that is, the activity of the hematopoiesis mutational signature– per year of age detected in patients with an identified mutation in a CH-related gene is significantly greater than that detected across patients without an identifiable CH-related mutation (Fig. 4f; Supp. Fig. 5c). The explanation for this finding is that hematopoietic mutations are more likely to appear above the threshold of detection of bulk sequencing the greater the CH clone. In samples carrying a mutation in a bona fide CH driver, it is more likely that this clone has expanded enough to identify a set of hitchhiking mutations through the reverse calling. Conversely, among samples without a CH-related mutation it is more likely that the clone is smaller or not present at all (detected hematopoietic mutations may be false positives of the reverse calling). In either case, the number of identified mutations is expected to be smaller across these patients.

We then set out to detect all CH cases across the metastasis (Fig. 4g) and primary cohorts (Supp. Fig. 5c–e). First, we determined that 141 CH cases in the metastasis cohort would be detected just by identifying somatic variants affecting genes in the list of 15 known CH drivers on the bases of the blood germline calling (4% of the patients in the metastasis cohort).

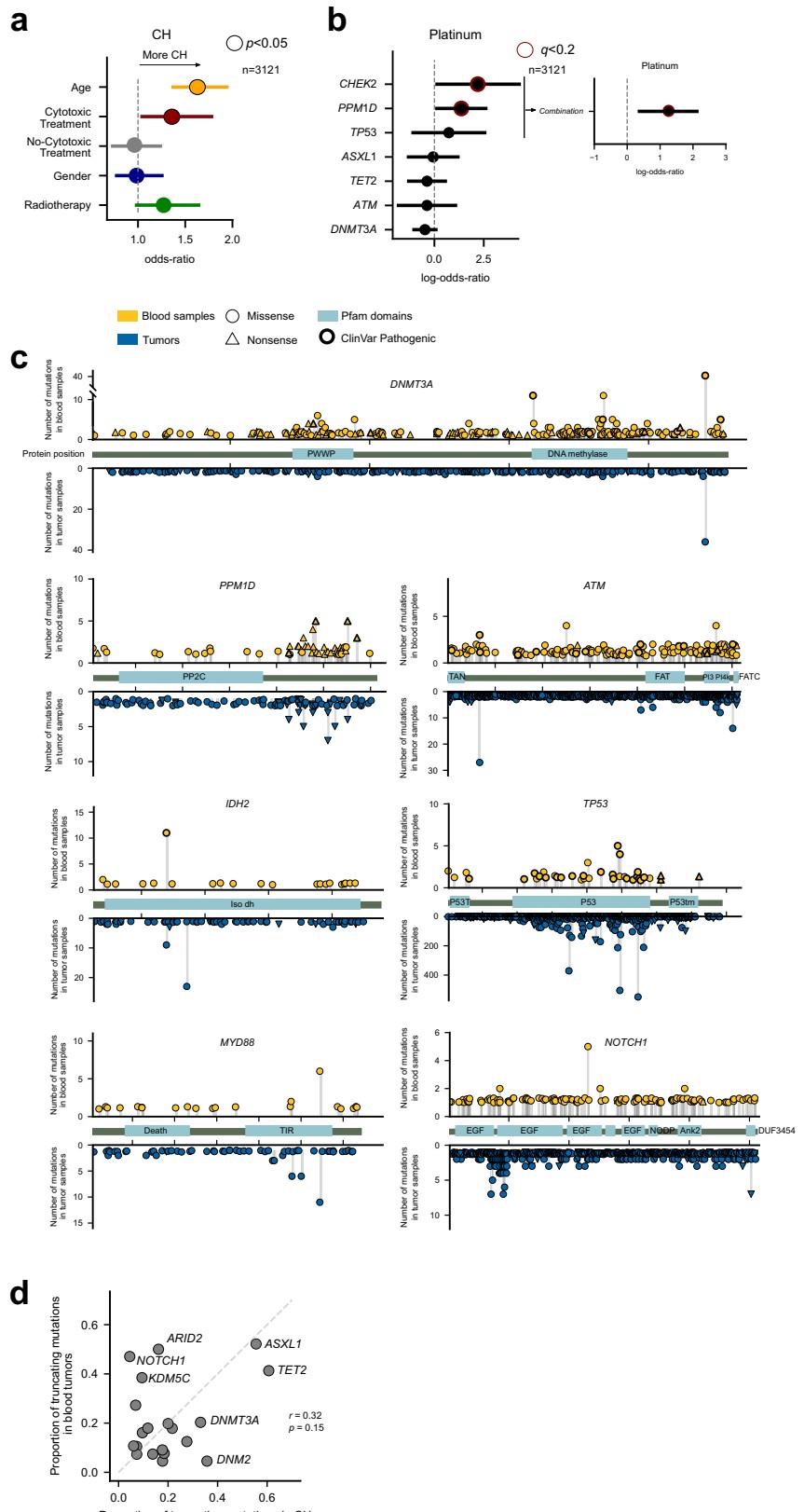

Using the reverse calling to identify somatic variants affecting these genes would add 99 CH cases (ascending to 7% of the total number). The addition of all CH-related genes to the compendium in this paper identifies 110 further CH cases (up to 10%), with 59 (ascending to 11%) more added if the set of CH driver genes identified across the targeted cohort is also considered.

Across donors in the primary cohort, 27% are detected as CH cases following the same criteria (Supp. Fig. 5d,e).

Finally, we assumed that any sample with a rate of hematopoiesis mutations per year above the median of the distribution of values observed for samples carrying CH-related mutations is a case of CH, even in the absence of identified driver

**Fig. 3 The drivers of clonal hematopoiesis. a** Logistic regression showing the relationship between several factors and the development of CH across 3121 donors with treatment annotation in the metastasis cohorts. For this analysis, a donor is considered to suffer CH if they bear a nonsilent mutation in a CH gene discovered in the analysis of the primary and/or metastasis cohorts. The age of the donors in these cohorts as well as their prior exposure to cytotoxic therapies significantly increase their likelihood of presenting clonal hematopoiesis. The bars represent the 95% confidence interval of the regression coefficients. *P*-values correspond to the results of the logistic regression. **b** Logistic regression showing the relationship between the presence of mutations in several genes and the prior exposure of donors in the metastasis cohort to platinum-based therapies across 3121 donors with treatment annotation in the metastasis cohort. Mutations in *CHEK2* and *PPM1D* are significantly more likely detected across platinum-exposed donors. The bars represent the 95% confidence interval of the regression coefficients. *P*-values correspond to the results of the logistic regression corrected by multiple tests carried out separately for different treatments. **c** Distribution of blood somatic mutations affecting seven genes selected from the CH drivers compendium across donors of the primary and metastasis cohorts (above the horizontal axis) in comparison to those observed in the same genes across 28076 tumors analyzed by the IntOGen resource[25] (below the horizontal axis). **d** Relationship between the fraction of truncating variants identified in genes with 10 or more mutations across blood samples in the primary and metastasis cohorts and across several cohorts of tumors[25]. The mutations in tumor samples have been obtained from the IntOGen resource. The *p*-value corresponds to the Pearson's correlation coefficient. Source data for panels **a**, **b**, **c**, and **d** are provided as Source Data files.

mutation (Fig. 4f). Thus, 562 (totalling 15%) blood samples in the metastasis cohort with no detectable CH-related mutations exhibit a rate of hematopoietic mutations comparable to that of samples with a mutation in a bona fide CH driver (Fig. 4g). We reasoned that at least some of these CH cases–with an appreciable clonal expansion–could be driven by mutations affecting yet unidentified CH drivers or may have resulted from expansion of HSCs due to non-genetic mechanisms.

Still, some CH cases may be driven by noncoding mutations. Whole-genome sequenced blood samples could in principle be employed to identify such non-coding driver mutations. This is not an easy task, as demonstrated by the search for non-coding cancer driver events[55,56]. The possibility is nevertheless opened by the reverse calling demonstrated here to set out to identify signals of positive selection in the observed pattern of mutations of different non-coding genomic elements. This is demonstrated with the results of OncodriveFML[42], MutSigCV_NC[56] and DriverPower[57] on mosaic mutations in non-coding genomic elements (Supp. Fig. 6a and Supp. Data file 3). The results of such analyses need to undergo a rigorous vetting process, as the distribution of mutations under neutrality in non-coding regions is still very difficult to model[58]. Alternatively, the functional effect of mutations overlapping particular non-coding regulatory elements, such as the binding site of a transcription factor in an enhancer element, may be assessed. For example, Supplementary Figure 6b illustrates the potential disruption of a binding site for *RARA* in an enhancer element regulating *TET2* according to geneHancer[59]. Supplementary Figure 6c (see more examples in Supp. Data file 4) presents the potential creation of a *SALL4* binding site in an enhancer regulating the expression of *GNAS*.

## Discussion

The extent of CH across patients with no known hematologic phenotype is currently not well gauged, although population studies have revealed that it is probably higher than anticipated a few years ago[2,3,13,14,16,21,22]. Understanding this extent and comprehensively identifying CH across healthy individuals is key to predicting potential future health hazards. One stepping stone in this path is the identification of all genes with mutations capable of driving CH. Moreover, the identification of all CH-related genes is a requisite to understanding the mechanisms behind this process and its relationship with disease conditions, as has been done for mutations affecting chromatin remodelling and DNA damage response genes classically associated with the condition[2,16,17,53]. In this regard, the discovery of CH-related genes across populations of various ethnicities and with different lifestyles, will allow us to understand the different constraints faced by hematopoietic cells in their evolution.

The main contribution of this work to the study of CH is the demonstration that cancer donor cohorts may be successfully repurposed–using tools developed for cancer genomics–to unbiasedly identify CH driver genes. First, we demonstrate that the existence of a second non-blood sample of the same donor refines the identification of somatic mutations in a blood sample, even if this is sequenced at low depth. The reverse calling implemented and tested here identifies blood somatic mutations with more sensitivity (across all discovery CH drivers) and more specificity (owing to the tumor paired sample) than a regular germline calling on a single blood sample, as done by previous studies[22]. (Importantly, the identification of mutational signatures active in a blood sample that may be the result of sequencing artifacts calls to caution when interpreting these blood mutations.) Second, we show that CH-related genes may be systematically and unbiasedly identified through the repurposing of tools aimed at identifying genes under positive selection in tumorigenesis.

The compendium of CH drivers that the combination of these two elements brings within reach will improve the identification of CH across healthy individuals. Importantly, some CH cases may be driven by larger chromosomal events, such as copy number changes, rather than by (or in addition to) point mutations[60]. While the size of the cohorts employed here limits the power of the discovery of CH drivers, and the mechanistic inferences that can be made from them, we envision that the application of this rationale to large tumor sequencing cohorts will contribute to expanding the list of CH drivers. This effort would benefit–as is apparent from the previous paragraph–from deeper sequencing of the reference blood samples in cancer genomics studies. Moreover, the evidence that CH may be present in a substantial number of samples in the absence of mutations of genes in the compendium underlines the pressing need to extend the discovery of CH drivers. In this regard, an analysis that repurposes many more tumor/blood paired samples obtained in the context of cancer genomics projects following the approach demonstrated in this paper is of paramount importance.

The experimental validation of the mutations observed in the genes of the compendium is out of the scope of this work. Nevertheless, before the compendium of mutational CH drivers may be translated into epidemiological studies and, in particular, into interventions aimed at preventing the effects of CH, the implications of mutations affecting CH driver candidates need to be established through combinations of in vitro, in vivo and population studies.

One clear benefit of a compendium produced via a systematic driver discovery effort with respect to the identification of recurrently mutated suspicious genes is that it will consider only those with clear signals of positive selection. Therefore, mutated

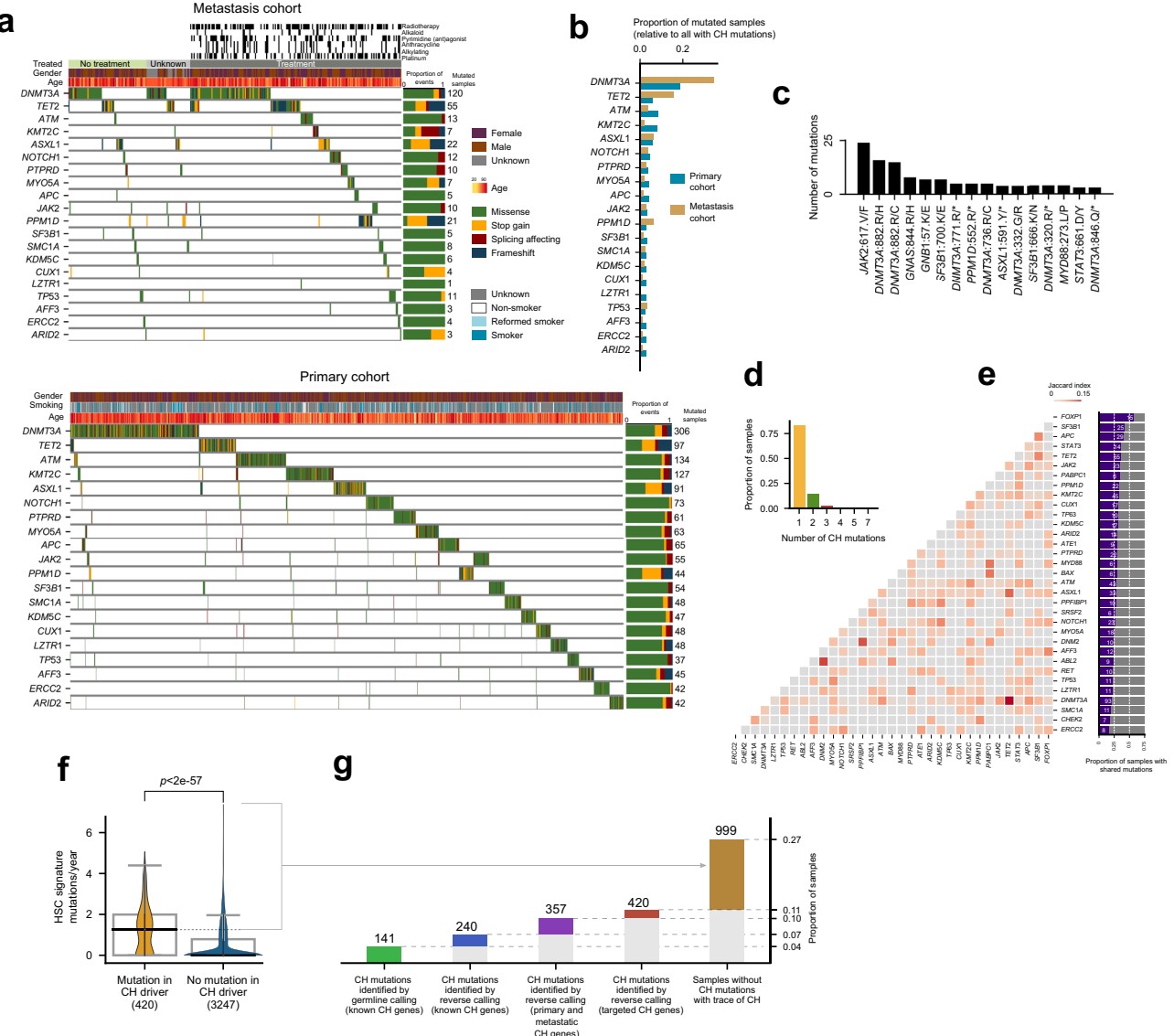

**Fig. 4 Clonal hematopoiesis across 12,000 donors. a** Blood somatic mutations in the 20 most recurrently mutated genes in the compendium across the metastasis (top) and primary (bottom) cohorts. **b** Frequency of mutation of CH drivers across the metastasis and primary cohorts. **c** The 16 most recurrently mutated hotspots in genes in the CH drivers compendium. **d** Number of donors in the two cohorts with mutations in genes in one or more CH drivers. **e** Frequency of co-occurring mutations in genes in the CH drivers compendium. Left, Jaccard's index; right, frequency of gene pairs co-mutation. **f** Distribution of the rate of hematopoietic mosaic mutations per year (total number of HSC mutations divided by age) across (left) donors bearing a mutation in genes in the CH drivers compendium ($N = 420$) and (right) donors with no detected mutations in any of these genes ($N = 3,247$). The horizontal dashed line extends out of the median of the distribution of rate of mutation per year of age of the donors with mutations in at least one CH gene, representing the donors in the second group that are considered to be cases of clonal hematopoiesis (see next panel). In the boxplots, the box represents the second and third quartiles, separated by a line indicating the median; the whiskers represent the minimum and maximum of the distribution excluding outliers. The two distributions were compared using the two-tailed Wilcoxon-Mann-Whitney test. **g** Number of donors (above the bars) in the metastasis cohort with clonal hematopoiesis recognizable using different criteria (cumulative bars). First, donors with mutations (detected in the germline calling) in any of the 15 known CH genes; second, donors with variants in known CH genes identified in reverse calling; third, donors with mutations in CH genes discovered across the primary or metastasis cohorts; fourth, donors with mutations in CH genes discovered in the targeted cohort; fifth, donors with no mutation in any gene within the compendium of CH drivers, but with more hematopoiesis mutations per year of age of the donor than the median rate of hematopoiesis mutations across donors in the four previous groups. Source data for panels **a**, **b**, **c**, **d**, **e**, **f** and **g** are provided as Source Data files.

genes that are passengers to the CH process will not be considered, even if they are known to be involved in tumorigenesis in solid tissues (see examples in Supp. Note). This, in turn, will result in a more accurate identification of CH cases across donors.

Although a set of CH-genes common to both cohorts is apparent from the discovery, a plethora of genes specific to each of them also appears. This is probably due to differences in both cohorts: primary vs metastatic tumors, with many donors in the latter having been exposed to chemotherapies. Mutations in some CH-related genes are indeed known to provide an advantage to hematopoietic cells under exposure to certain cytotoxic treatments. Other aspects, such as the different composition of both cohorts, in terms of human populations and tumor types represented may also have a bearing on the differences in CH-related genes discovered in each[61]. Further studies are needed to clarify this point, which the availability of the discovery presented here

now makes possible to undertake. Importantly, the fact that some CH genes reported in the compendium are not common across hematopoietic malignancies suggests that at least in some cases, CH and hematopoietic tumors may present totally different evolutionary paths. Still, CH cases underpinned by mutations in these genes may have known (e.g., cardiovascular disease or hypertension) or novel long-term effects on the health of carriers.

The unbiased snapshot of the compendium of CH drivers identified has a series of implications for both CH and cancer research. It may be directly employed in the research of the molecular mechanisms underlying CH in different scenarios. The list of 64 genes discovered can also be employed to refine the identification of the condition across human donors. Such donor-wise identification of CH would require the analysis of a single blood sample, identifying variants affecting the genes in the compendium. An important warning arising from this work is that not all blood mutations affecting cancer driver genes play a role in CH. Thus, the results from sequencing panels that include genes without signals of positive selection in CH need to be carefully interpreted. In the cancer research field, our results support the idea that sequencing cell-free DNA isolated from blood samples with the aim of identifying tumor mutations in circulating genetic material may produce false-positive results caused by the detection of CH mutations[62,63].

Whereas the compendium of CH drivers is a prerequisite for the detection of CH across individuals, a second necessary step consists in evaluating the capability of individual mutations in CH drivers to provide a selective advantage to HSCs. If only mutations with experimentally validated effect on CH or identified through epidemiological studies are considered as CH drivers, the prevalence of CH is underestimated. On the other hand, taking into consideration all mutations affecting CH drivers probably leads to an overestimation of CH. We envision that the approach of in silico saturation mutagenesis of genes involved in tumorigenesis recently developed by us will become useful in this task[64].

## Methods

**Sequences of samples from the primary and metastasis cohorts**. The sequences of solid tumors and their paired blood samples (BAM files) were obtained from the Genomic Data Commons (GDC; https://portal.gdc.cancer.gov[65]) portal upon dbGAP request (phs000178.v11.p8 dataset; https://www.ncbi.nlm.nih.gov/projects/gap/cgi-bin/study.cgi?study_id=phs000178.v11.p8) for the primary cohort ($N = 8530$) and from the Hartwig Medical Foundation (HMF; https://www.hartwigmedicalfoundation.nl[29]) repository, upon request to HMF for the metastatic cohort ($N = 3785$).

**HMF gemline calling**. The germline variant calls carried out using the HaplotypeCaller[66] for the metastasis cohort were obtained as part of the HMF dataset[29]. All mutations, independently of the quality filters, were used to compare the sensitivity of this germline calling with the reverse calling developed in the paper (see below). This produces very conservative estimations.

**Detecting somatic mutations in blood samples across the primary and metastasis cohort (reverse calling)**. The variant calling was carried out using the Google Cloud Platform (metastasis cohort) and our in-house computer cluster (primary cohort). Briefly, the matched blood and tumoral BAM files–masked and deduplicated using GATK[66]–of 3785 whole-genome samples (metastasis cohort) and 8530 whole-exome sequenced samples (primary cohort) were obtained as described above. The variant calling was carried out using Strelka2[31] (employing default parameters) with the blood sample as the tumoral input and the tumor sample as control (reverse calling). In the case of patients with more than one tumor sample, one of them was randomly selected and included in the calling. All variants with two or more supporting reads matching the caller PASS filter and with VAF < 0.5 were kept. Mutations in lowly mappable regions as defined by the DUST algorithm[67] (k = 30) and UMAP[68] (36-kmers) were excluded. Contiguous variants were merged into double-base substitutions. Variants with greater frequency across the cohorts than the *DNMT3A* R882H or *JAK2* V617F hotspot in a cohort-specific Panel of Normals, or PoN (obtained from GDC and HMF for the primary and metastasis cohorts, respectively) and in gnomAD[69] v2.1 were removed. This was equivalent to discarding variants present in these datasets with a

minor allele frequency greater than 0.002 in PoN TCGA, 0.008 in PoN HMF and 0.0003 in gnomAD v2.1. Additionally, common SNPs defined by the snp151Common UCSC track[70] and dbSNP[71] were excluded. Mutations within segmental duplications, simple repeats and masked regions as defined in UCSC tracks were also removed. Finally, samples with the mutation count above the 97.5 percentile of the mutation burden across the cohort were deemed unreliable and excluded for further analyses. We call the set of variants obtained after the application of these filters the **full set**.

Two more conservative subsets were generated from the full set in the primary and metastasis cohorts. The first (**mutect set**) comprises only variants that were also identified by Mutect2[32] (only for the metastasis cohort). Mutect2 was executed with the following parameters:

gatk --java-options "-Xmx4g" Mutect2 -R {} -I {} -tumor {} -I {} -normal {} --germline-resource {} --panel-of-normals {} -L {} -O {} --QUIET

The MergeMutectStats and FilterMutectCalls commands were then run subsequently.

Second, we applied MosaicForecast (https://github.com/parklab/MosaicForecast v.0.0.1)[34], a software designed to phase mutations to polymorphisms with the aim of identifying somatic mutations with very low VAF and also of predicting mosaicism for the unphased ones with a random forest classifier. As a result, we obtained a subset of mosaic-phased mutations, and a subset of mutations likely to be somatic (**mosaic set**). In the primary cohort, only the mosaic set was obtained through filtering of the full set.

**Blood somatic mutations in targeted-sequenced samples**. Somatic blood mutations identified across 24,146 targeted-sequenced blood samples[17] were directly obtained from cBioportal (https://www.cbioportal.org/)[72].

**Detection of mutational signatures**. To identify mutational signatures active in the metastasis cohort, we employed the mosaic set and applied a non-negative matrix factorization approach[73], using the SigProfilerJulia (bitbucket.org/bbglab/sigprofilerjulia) implementation prepared in our lab[74] of the algorithm developed by Alexandrov et al.[73]. Only samples with more than 100 mutations were included in the analysis. The resulting signatures were then compared to the PCAWG COSMIC V3[36] catalog using the cosine similarity measure. No signature was extracted from the mutations identified in the primary (exome-sequenced) cohort due to their low numbers.

Whole-genome somatic variants of 23 blood samples from healthy donors of different ages were obtained upon request to the authors of Osorio et al.[35]. The Hematopoietic Stem Cell Signature (HSC signature)[35] was computed as the average number of mutations observed across the 23 healthy blood samples in each of the 96 tri-nucleotide channels normalized by the total number of mutations observed.

**Discovering the compendium of CH driver genes**. The discovery of genes with signals of positive selection was carried out using the IntOGen pipeline[25]. Briefly, the IntOGen pipeline implements seven complementary methods to identify signals of positive selection in the mutational pattern of genes and integrates their outputs. The pipeline first pre-processes the somatic mutations across samples to filter out hypermutator samples, map all mutations to the GRCh38 assembly of the human genome and retrieve information necessary for the operation of the seven driver detection methods. Then, the methods are executed and their outputs combined using a weighted voting approach with weights adjusted depending on the credibility awarded to each method. Finally, in a post-processing step, spurious genes that result from known artifacts are automatically filtered out (see Supp. Note 1). The version of the pipeline used in this study is described at length at www.intogen.org/faq and in *Martinez-Jimenez* et al.[25].

The IntOGen pipeline was run on the full set, the mutect set (metastasis cohort) and the mosaic set of mutations independently. Subsequently, genes that were identified as having signals of positive selection only in the full set were required to possess extra evidence (either identified by the pipeline run on a filtered set, or included within the Cancer Gene Census[50]) to be included in the final list. To compare CH-related genes according to this unbiased discovery to the prior knowledge on the genetics of this process, we used i) a list of genes involved in CH (ground truth of known CH genes[34]), ii) genes known to drive myeloid malignancies[17,21], and iii) all genes annotated in the Cancer Gene Census[50].

Only a subset of the methods (capable of building a background mutations model from the segment of the exome probed in the panel) were run on the set of somatic mutations identified in the blood samples of the targeted cohort. OncodriveCLUSTL, OncodriveFML, dNdScv (without genome-wide mutation rate covariates, as in ref.[75]), and HotMaps were run through the IntOGen pipeline, and their individual outputs collected. Significant genes (with a FDR cutoff of 0.01) in the analysis of any method (that is, a union of the lists) were considered CH drivers in this cohort.

The final snapshot of the compendium of CH driver genes was integrated by the union of the lists of genes identified across the three cohorts.

**Identification of blood samples with clonal hematopoiesis**. To identify individual donors in the metastasis cohort with clonal hematopoiesis, we considered all mutations that putatively affected the protein sequence of any gene discovered as CH-related across the cohort in the present study (separated in the different

categories presented in Fig. 4g). We then computed the median rate of hematopoiesis mutations per year of age across the blood samples of these donors. All donors with no mutation in a discovery CH gene but with a rate of hematopoiesis mutations per year of age greater than this median value were also considered as CH cases (the final group in Fig. 4g).

**Logistic regressions**. Inspired in a previous work[17], we used multivariable logistic regression to assess the association between clonal hematopoiesis and therapy, age and gender. We also used it to compute the association between mutations in specific genes (or groups thereof) and the exposure of donors to specific chemotherapeutic drugs. Multiple test correction (Benjamini-Hochberg FDR) was used for gene-specific analyses.

**Identifying expressed CH-related genes**. We computed the distribution of the expression of each gene across the expression of bone marrow CD34 + cells obtained from The Gene Expression Omnibus (GSE96811[76]). These cells are phenotypically the closest to the HSCs. We deemed a gene expressed across the cells when the maximum value of its distribution was above 15 fpkm.

**Comparison of blood somatic mutations with tumor mutations**. The distribution of mutations in CH driver genes observed across blood samples from the primary and metastasis cohorts was compared to that observed across hematopoietic malignancies in IntOGen[25]. ClinVar pathogenic and likely pathogenic variants were obtained from ref. [77].

**Non-coding blood somatic variants in CH**. Three state-of-the-art methods designed to detect positive selection in the mutational patterns of non-coding genomic elements (OncodriveFML[42], DriverPower[57], MutSigCV_NC[55]) were run with default parameters. The non-coding genomic elements were obtained from the PanCancer Analysis of Whole Genomes (PCAWG)[26]. A FDR cutoff of 0.2 was applied.

The set of transcription factor (TF) binding motifs was obtained from ref. [78]. Models with A,B and C qualities were kept. Only TF expressed in CD34 + cells according to GSE96811 were allowed in the analysis. H3K27ac ChIP data for CD34 + samples was obtained from ENCODE[79]. Enhancer element coordinates, as well as their defined target genes, were retrieved from geneHancer[59] via the UCSC genome browser. Briefly, mutations intersecting with H3K27ac peaks and an enhancer defined by geneHancer were expanded 15 bp upstream and downstream. Then, using FIMO[80] the binding affinity of these sequences was determined for both the mutant and the reference allele. When the significance of the binding was less than 0.0001 in the reference but not in the mutant, we labeled the instance as disruption (and creation, if the case is the opposite). We retained only results for which the gene closest to the disruption/creation of a TF is a CH driver. Visualization of the genomic context of the mutations represented in Supplementary Figure 6 was performed using pyGenomeTracks[81].

**Reporting summary**. Further information on research design is available in the Nature Research Reporting Summary linked to this article.

## Data availability

The sequencing data to carry out the reverse calling of blood somatic mutations (and germline variants across donors) is available via dbGaP (TCGA; phs000178.v11.p8) and HMF (https://hartwigmedical.github.io/documentation/data-access-request-application.html, version DR110). Access to these protected data must be requested from TCGA and HMF. The procedure and conditions to access these datasets are detailed in the sites referenced above. Gene expression in bone marrow CD34 + cells are available at The Gene Expression Omnibus (GSE96811). H3K27ac ChIP data for CD34 + samples are available from ENCODE ([https://www.encodeproject.org/experiments/ENCSR891KSP/]). Mutations in CH drivers across hematopoietic malignancies are available from IntOGen [http://www.intogen.org/ch]. Disease-related variants are available from ClinVar [https://ftp.ncbi.nlm.nih.gov/pub/clinvar/]. We have prepared flat files containing the set of blood somatic mutations identified in both datasets and have made them available through HMF and dbGaP following the same procedure to access the original datasets. HMF blood somatic mutations are available as part of the data access request to HMF (see above). TCGA blood somatic mutations are available through dbGaP (phs002867) to researchers who have obtained permission to access protected TCGA data. Panel-sequenced data from the IMPACT targeted cohort is available through cBioPortal ([https://www.cbioportal.org/study/summary?id=msk_ch_2020]). The compendium of CH drivers is available via www.intogen.org/ch. Other datasets employed in specific analyses are described in prior sections of these Methods and in README files within the code repository. Source data are provided with this paper.

## Code availability

The programs required for the variant calling are all open source, as is the IntOGen pipeline (available at www.intogen.org), and the programs used in the analysis of CH

non-coding mutations (listed in the previous section). All other analyses described in the paper were implemented ad hoc in Python. A code repository has been prepared with scripts employed in the reverse calling and jupyter notebooks required to reproduce all downstream analyses and the figures of the paper. This repository is available at [https://github.com/bbglab/ch-drivers]. A Zenodo repository pointing to this code repository has also been set up[82] (doi: 10.5281/zenodo.6521953).

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

## Acknowledgements

The authors wish to thank fruitful discussions of the results of the paper with Jose J. Fuster, on potential caveats of the reverse calling approach with Santiago Gonzalez. N.L-B. acknowledges funding from the European Research Council (consolidator grant 682398) and ERDF/Spanish Ministry of Science, Innovation and Universities - Spanish State Research Agency/DamReMap Project (RTI2018-094095-B-I00) and Asociación Española Contra el Cáncer (AECC) (GC16173697BIGA). IRB Barcelona is a recipient of a Severo Ochoa Centre of Excellence Award from the Spanish Ministry of Economy and Competitiveness (MINECO; Government of Spain) and is supported by CERCA (Generalitat de Catalunya). O.P. is the recipient of a BIST PhD fellowship supported by the Secretariat for Universities and Research of the Ministry of Business and Knowledge of the Government of Catalonia, and the Barcelona Institute of Science and Technology (BIST). This publication and the underlying research are partly facilitated by Hartwig Medical Foundation and the Center for Personalized Cancer Treatment (CPCT) which have generated, analyzed and made available data for this research. We would like to thank Paul Wolfe from Hartwig Medical Foundation for his guidance in the Google Cloud Platform usage. This publication and the underlying research are also partly facilitated by data collected and made public by The Cancer Genome Atlas network.

## Author contributions

O.P., A.G.-P. and N.L.-B. designed the project. O.P. carried out all the analyses and prepared the figures. O.P and I.R-S implemented the pipeline in GCP. I.R-S implemented the web site. N.L.-B. and A.G.-P drafted the manuscript. O.P., A.G.-P. and N.L.-B. edited the manuscript. A.G.-P. and N.L.-B. supervised the project.

## Competing interests

The authors declare no competing interests.
