## [Peer Review File · Nature Communications]

nature portfolio

Peer Review FileEditorial Note: This manuscript has been previously reviewed at another journal that is not operating a transparent peer review scheme. This document only contains reviewer comments and rebuttal letters for versions considered at Nature Communications.

REVIEWERS' COMMENTS

Reviewer #2 (Remarks to the Author):

I thank the authors for their careful replies to my questions and have no more comments.

Reviewer #3 (Remarks to the Author):

The authors have addressed all the concerns I raised when I reviewed a previous version of the manuscript for a different journal. I have no further comment.

Reviewer #4 (Remarks to the Author): Expert in clonal haematopoiesis and genomics

In the current work Pich et al. searched for somatic mutations in the blood of 12000 whole exome/whole genomes cancer patients.

As they claim in the discussion: " The main novel contribution of this work to the study of CH is the demonstration that cancer donor cohorts may be successfully repurposed --using tools developed for cancer genomics to unbiasedly identify CH driver genes. First, we demonstrate that the existence of a second non-blood sample of the same donor refines the identification of somatic mutations in a blood sample, even though this is sequenced at low depth."

Major concern:

1. The approach the authors propose is not novel and an almost identical approach was used by Xie et.al Nat Med 2014. Here is what Xie wrote in 2014 9 (doi:10.1038/nm.3733): "The collection of both tumor and matched blood normal exome data by TCGA provides a unique comparative resource for identifying those somatic variants in blood that contribute to clonal expansion".

I do admit that the Xie manuscript included only 2728 samples and only whole exomes. However the approach is not novel.

It should be noted that many of the possibly novel variants in Xie et.al (Supplementary Table 7) like in the genes SOS1 SNX25 and even ASXL2 were never confirmed as CH genes. Altogether one should be very careful while claiming about new CH genes, as many sequencing efforts have been done, and a very detailed validation is needed.

The second claim by the authors is that: " Second, we show that CH-related genes may be systematically and unbiasedly identified through the repurposing of tools aimed at identifying genes under positive selection in tumorigenesis."

Major concern

2. At least one of the variants with suspected positive selection should be validated in an experimental way, and should be analyzed in different leukemia studies. As it can cause AML, MPN, CLL, and maybe others. If it can be found in one of these diseases, it is not novel player but rather point to a latent phase before diagnosis.

Dear Editor,

We are delighted that our work has been considered for publication, based on its new evaluation by the reviewers. Please, find below our responses to their comments on the latest version of our manuscript.

REVIEWERS' COMMENTS

Reviewer #2 (Remarks to the Author):

I thank the authors for their careful replies to my questions and have no more comments.

We thank the reviewer for their evaluation of our manuscript.

Reviewer #3 (Remarks to the Author):

The authors have addressed all the concerns I raised when I reviewed a previous version of the manuscript for a different journal. I have no further comment.

We thank the reviewer for their appreciation of our work.

Reviewer #4 (Remarks to the Author): Expert in clonal haematopoiesis and genomics

In the current work Pich et al. searched for somatic mutations in the blood of 12000 whole exome/whole genomes cancer patients.

As they claim in the discussion: " The main novel contribution of this work to the study of CH is the demonstration that cancer donor cohorts may be successfully repurposed --using tools developed for cancer genomics to unbiasedly identify CH driver genes. First, we demonstrate that the existence of a second non-blood sample of the same donor refines the identification of somatic mutations in a blood sample, even though this is sequenced at low depth."

We thank the reviewer for their assessment of our work.

Major concern:

1. The approach the authors propose is not novel and an almost identical approach was used by Xie et.al Nat Med 2014. Here is what Xie wrote in 2014 9 (doi:10.1038/nm.3733): "The collection of both tumor and matched blood normal exome data by TCGA provides a unique comparative resource for identifying those somatic variants in blood that contribute to clonal expansion".

I do admit that the Xie manuscript included only 2728 samples and only whole exomes. However the approach is not novel.

We thank the reviewer for pointing this out. While the study by Xie *et.al*/ Nat Med 2014 (ref. 22 in our manuscript) did seek to identify clonal hematopoiesis related variants across blood samples from TCGA, it did so through a one-sample germline mutation calling. Although the authors compared the variants obtained through this germline mutation calling with somatic mutations observed in the tumors, they did not exploit the second (tumor) sample available from these patients in a reverse calling approach as the one implemented in our study.

We have modified a sentence of the Discussion section to more clearly acknowledge this precedent.

The reverse calling implemented and tested here identifies blood somatic mutations with more sensitivity (across all discovery CH drivers) and more specificity (owing to the tumor paired sample) than a regular germline calling on a single blood sample, as done by previous studies exploiting solely blood samples from tumor patients cohorts.

It should be noted that many of the possibly novel variants in Xie *et.al* (Supplementary Table 7) like in the genes SOS1 SNX25 and even ASXL2 were never conformed as CH genes. Altogether one should be very careful while claiming about new CH genes, as many sequencing efforts have been done, and a very detailed validation is needed.

We agree with the reviewer that the novel genes discovered in our study as part of the compendium of CH drivers require (and merit) careful validation.

The second claim by the authors is that: "Second, we show that CH-related genes may be systematically and unbiasedly identified through the repurposing of tools aimed at identifying genes under positive selection in tumorigenesis."

Major concern

2. At least one of the variants with suspected positive selection should be validated in an experimental way, and should be analyzed in different leukemia studies. As it can cause AML, MPN, CLL, and maybe others. If it can be found in one of these diseases, it is not novel player but rather point to a latent phase before diagnosis.

As stated above, we agree with the reviewer on the need of validation of the mutations observed in the genes in the CH compendium. Nevertheless, a thorough validation –which should extend beyond one single experiment supporting the functionality of one variant– lies outside the scope of our study.

It is also important to mention that we discuss this issue with more detail in the Supplementary Note.

Following the reviewer's comment, we now clearly state this in the Discussion section.

The experimental validation of the mutations observed in the genes of the compendium is out of the scope of this work. Nevertheless, before the compendium of mutational CH drivers may be

translated into epidemiological-driven studies and, in particular, into interventions aimed at preventing the effects of CH, the implications of mutations affecting novel CH driver candidates need to be established through combinations of in vitro, in vivo and population studies.